# Recombinant 60-kDa heat shock protein from *Paracoccidioides brasiliensis* induces the death of mouse lymphocytes in a mechanism dependent on Toll-like receptor 4 and tumor necrosis factor

Igor Emiliano L. Souza[1], Fabrício F. Fernandes[2], Ademilson Panunto-Castelo [1,3]*

1 Graduate Program in Basic and Applied Immunology, Ribeirão Preto Medical School, University of São Paulo, Ribeirão Preto, São Paulo, Brazil, 2 Federal Institute of Education, Science and Technology of Mato Grosso, Campus Sorriso, Sorriso, State of Mato Grosso, Brazil, 3 Department of Biology, Faculty of Philosophy, Sciences and Letters at Ribeirão Preto, University of São Paulo, Ribeirão Preto, São Paulo, Brazil

* apcastelo@usp.br

**Data Availability Statement:** All relevant data are within the paper.

## Abstract

*Paracoccidioides* fungi are thermodimorphic microorganisms that cause paracoccidioidomycosis (PCM), an autochthonous disease from Latin America, with most cases in Brazil. Humans become infected by inhaling conidia or mycelial fragments that transform into yeast at body temperature. These fungi cause chronic-granulomatous inflammation, which may promote fibrosis and parenchyma destruction in the lungs. In response to stress imposed by the host, fungi *Paracoccidioides* spp. increase the expression of heat shock proteins (HSP), which protect them by sustaining cellular proteostasis. Our group has studied the role of HSP60 in PCM, and previous data show that the recombinant HSP60 (rHSP60) has a deleterious effect when used in a single dose as therapy for experimental PCM. Here, we investigated the mechanism by which rHSP60 could worsen the disease. We found that rHSP60 caused the viability loss of splenic or lymph node cells from both immunized and non-immunized mice, including in splenic T lymphocytes under polyclonal stimulation with concanavalin A, probably by undergoing apoptosis. Among analyzed splenic cells, lymphocytes were indeed the main cells to die. When we investigated the death mechanisms, remarkably, we found that there was no viability loss in rHSP60-stimulated splenic cells from mice deficient in Toll-like receptor 4, TRIF adapter protein, and TNF receptor 1(TNFR1), as well as rHSP60-stimulated WT cells incubated with anti-TNF antibody. Besides, caspase-8 inhibitor IETD-CHO blocked the rHSP60 effect on splenic cells, suggesting that rHSP60 induces the extrinsic apoptosis pathway dependent on signaling via TLR4/TRIF and TNFR1.

## Introduction

*Paracoccidioides* is a genus of fungi that cause paracoccidioidomycosis (PCM), a systemic disease characterized by a chronic granulomatous inflammation that affects mainly the lungs, and

**Funding:** The São Paulo Research Foundation [grant numbers 2017/01390-8, and 2021/01528-5], and CAPES.

about 85% of registered cases have occurred in Brazil [1–3]. At the beginning of the infection, *Paracoccidioides* spp. need to adapt to the host tissue conditions, which rapidly becomes an inflammatory microenvironment [4]. One of the fungal adaptations consists of the morphologic transition from hyphae to yeast, i.e., from the saprophytic and infective form to the parasitic one, making the fungus more resistant to the components of innate immunity [1].

Among the components in *Paracoccidioides* spp. that are expressed in response to changes in temperature and other stressful conditions are the heat shock proteins (HSPs) [5]. HSPs are a class of highly conserved molecules with constitutive expression in cells of all living beings [6, 7]. Usually, HSPs can be found inside the cell in different compartments, such as cytosol, mitochondria, chloroplasts, and nucleus [8, 9]. They can be overexpressed in the cells under stress conditions, e.g., high temperature, hypoxia, and exposure to heavy metals or other toxic agents [10]. The physiological role of HSP correlates with sophisticated mechanisms, which allow the correct folding of peptides and avoid the formation of non-specific proteins [11]. While HSPs are essential for fungi and many other pathogenic microbes as a survival strategy in the host, the increase of some of these proteins promotes their recognition by host immune system [12].

In PCM studies, we and others have shown that an antibody response is developed against HSP in PCM patients since a significant percentage of sera from these patients reacted with recombinant 60-kDa (rHSP60) from *P. lutzii* [13] and *P. brasiliensis* [14]. Besides, we have also studied the use of rHSP60 from *P. brasiliensis* as a treatment for PCM on a murine model. Intriguingly, we observed that HSP60 and its recombinant counterpart had a harmful effect when administered in one single dose, inducing diffuse inflammation in the lungs and loose granulomas; nonetheless, the treatment with three doses decreased the fungal burden and lung inflammation [15, 16]. These results with three-dose treatment corroborated the Soares *et al.* study [17], which showed that the vaccination with three doses of rHSP60 induced protection against *Paracoccidioides* infection in mice. Although these results with three doses suggest that HSP60 may be important for the development of a protective immune response in the host, intriguingly the response with just one dose of rHSP60 worsened the infection. Therefore, in the present study, we sought to investigate the possible mechanism by which rHSP60 induced the detrimental effect on *P. brasiliensis*-infected mice.

## Materials and methods

### Mice

Six-to-eight-week-old male C57BL/6 (WT) mice were obtained from the Central Animal House of the University of São Paulo (USP), *Campus* Ribeirão Preto. Mutant mice for TRIF adaptor protein (TRIF$^{lps2/lps2}$) and deficient mice for TLR3 (TLR3$^{-/-}$), TLR4 (TLR4$^{-/-}$), myeloid differentiation factor 88 (MyD88$^{-/-}$), TNF receptor 1 (TNFR1$^{-/-}$), and TNF receptor 1 and 2 (TNFR1R2$^{-/-}$) with the same genetic background, age, and gender as the WT mice were purchased mice from the Special Mice Breeding Center of the Ribeirão Preto Medical School, USP (FMRP-USP). Mice were acclimated to the facility for one week before initiating the experiment, housed in individually ventilated microisolators, light-tight cabinets (Alesco, Capivari, Brazil), maintained at 20–22˚C, a 12 h light-dark cycle, and with access to chow and water ad libitum. We cleaned all microisolators twice a week and bedded them with autoclaved softwood shavings. Mice were maintained under specific pathogen-free conditions. For all immunization and euthanasia approaches, mice were previously anesthetized by intraperitoneal injection of ketamine (100 mg/kg) and xylazine (8 mg/kg) mixture. All mice experimental procedures were performed in accordance with the Guide for the Care and Use of Laboratory Animals of the National Research Council and approved by the Ethics Committee on Animal Use

of the Faculty of Philosophy, Sciences and Letters of Ribeirão Preto, USP, protocol 15.1.1770.59.0.

## Production of rHSP60

rHSP60 was expressed and purified as described by Fernandes *et al* [15]. Briefly, *E. coli* transformed with pET28a– *HSP60* vector were grown in LB medium supplemented with kanamycin sulfate (50 μg/mL). After inducing with 0.5 mM isopropyl-β-D-thiogalactopyranoside for 6 hours, bacterial cells were lysed by sonication in lysing buffer (50 mM $NaH_2PO_4$, 300 mM NaCl, 5 mM 2-mercaptoethanol, and 0.5% Triton X-100, pH 8.0). The sample was centrifuged at $7,000 \times g$ and the pellet washed 5 times with the lysing buffer. Pellet inclusion bodies was solubilized in a denaturing buffer (50 mM $NaH_2PO_4$, 300 mM NaCl, 30 mM imidazole, 7 M urea, 5 mM 2-mercaptoethanol, and 0.5% Tween 20, pH 8.0) for 1 hour. After centrifugation at $7,500 \times g$, the supernatant was clarified through 0.22 μm filter, and submitted to a $Ni^{2+}$–Sepharose affinity column (His-Trap; GE Healthcare). rHSP60 purified was concentrated and refolded by dialysis against phosphate-buffered saline (PBS) in a Amicon Ultra 15 device with a molecular weight cut-off of 10-kDa (Merck, Cork, Ireland). Protein concentration was determined using Quick Start Bradford Protein Assay (Bio-Rad, Hercules, USA). Purified rHSP60 sample was analyzed by SDS-PAGE. The sample contained less than 0.05 ng/mL of bacterial endotoxin, as determined by the Limulus amoebocyte lysate assay (Sigma-Aldrich, St. Louis, USA).

## Proliferation assays

To determine antigen-specific cell-mediated immune response, we used cells from drain popliteal lymph node (pLN) from mice immunized in the hind footpad with 50 μL of a preparation containing 50 μg of rHSP60 or ovalbumin (OVA) emulsified (v/v) in complete Freund adjuvant (CFA). The animals were anesthetized as described above and submitted to euthanasia by cervical dislocation 7 days after immunization. The pLN was aseptically removed and dissociated, and the cells were washed three times with PBS and stained with carboxyfluorescein diacetate (CFSE) to a final concentration of 1.25 mM. After 5 minutes at room temperature, inactivated fetal bovine serum (FBS) was added to the suspension in a 5% final concentration. Then the cells were washed with PBS and resuspended in complete RPMI medium (cRPMI), which was comprised of RPMI 1640 supplemented with 10% (v/v) of heat-inactivated fetal bovine serum (Life Technologies), 2 mM of L-glutamine, 100 U/mL penicillin, and 100 μg/mL streptomycin (all reagents from Life Technologies). Cell suspension at the density of $1 \times 10^6$ cells/mL were seeded in 24-well plates (Corning) and cultured without antigen (negative control) or pulsed with 2, 10, 25 or 50 μg/mL of rHSP60 for 48 hours. For OVA-stimulated pLN cells, of $1 \times 10^6$ cells/mL were cultured without stimulus or with only OVA at 50 μg/mL or with OVA added with 2, 10, and 50 μg/mL of rHSP60 for 48 hours. Cultures stimulated with 2 μg/mL of concanavalina A (ConA) were used as positive controls. Cultures were incubated for different times at 37°C under 5% $CO_2$. The cells were then washed with 0.5% bovine serum albumin (BSA) in PBS and analyzed by flow cytometry (Guava® easyCyte™ 8HT, Millipore). The analysis was performed at the Incyte program.

In some experiments, the effect of rHSP60 was evaluated in spleen cells from mice immunized with 3 doses of 50 μg of rHSP60 in intervals at 15 days or non-immunized (naive) mice. Briefly, animals were submitted to euthanasia as described above and had the spleen aseptically removed and dissociated. The cell suspension was treated with red blood cell lysing buffer (9 parts of 0.16 M ammonium chloride and 1 part of 0.17 M Tris-HCl, pH 7.5) for 4 minutes, washed three times with PBS and labeled with CFSE as described above. To evaluate the effect

of rHSP on B lymphocyte proliferation, after CFSE labeling, splenic cells were cultured with 25 μg/mL of LPS only or LPS plus 2, 10, or 50 μg/mL of rHSP60. The cultures and flow cytometry analysis were performed as described above for pLN cells. The concentration of nitrite ($NO_2^-$) in spleen cells was measured by a microplate Griess assay. The $NO_2^-$ concentration was determined using a standard curve of 1 to 200 μM NaNO2.

To analyze the effect of rHSP60 on cell subsets, the spleen cells at $4.5 \times 10^5$ cells/mL without labeling with CFSE were incubated with 25 or 50μg/mL of rHSP60 for 24 hours. Afterward, the cell suspensions were stained with phycoerythrin (PE)-conjugated anti-CD3, Allophycocyanin (APC)- conjugated anti-CD19, and peridinin chlorophyll protein–cyanine 5.5 (PerCP-Cy5.5)-conjugated anti-CD11b antibodies (purchased from all reagents from BD Biosciences, San Diego, USA). Despite the rHSP60 sample having an acceptable amount of LPS, we still added 30 μg/mL of polymyxin B (PMX, Sigma-Aldrich) in the cell culture. Besides, we digested 25 μg/mL of rHSP60 with 1 μg/mL of proteinase K (Invitrogen) for 5 hours at 37°C, following denaturation of the enzyme by heating at 95°C for 20 minutes. After cooling, the spleen cells stimulation was performed as described above. In these experiments, flow cytometry was carried out on the FACSCanto II Flow cytometer (BD Biosciences), and the analysis done using the FlowJo software (BD Biosciences).

## Detection of apoptosis

The cell cultures of spleen cells were stimulated for 24 hours with 25 μg/mL of rHSP60. Afterward, the cells were collected and washed twice with PBS. The cells were analyzed for cell death with the fluorescein isothiocyanate (FITC)-conjugated annexin V and propidium iodide (PI), according to the manufacturer's recommendations (Apoptosis Detection Kit, Sigma-Aldrich). The flow cytometry was carried out on the FACSCanto II Flow cytometer (BD Biosciences), and the analysis done using the FlowJo software (BD Biosciences).

## Cell viability assay

WT mice were injected intramuscularly (i.m.) once with 50 μg/mL of rHSP60 or thrice in 15-day intervals between doses. Seven days after the only or last dose of rHSP60, the animals were euthanized as described above and had the spleen removed. Likewise, spleen from non-immunized WT, TRIF[lps2/lps2], TLR3[-/-], TLR4[-/-] mice also was used. Fragments of the spleen were homogenized and passed through a mesh. After washing the cells with RPMI 1640, the erythrocytes were lysed with ACK buffer (150 mM $NH_4Cl$, 10 mM $KHCO_3$, 1 mM EDTA, pH 7.3) for 5 minutes at 4°C. Cell suspension was washed in PBS and resuspended in cRPMI at a concentration of $1 \times 10^6$ cells/m. To each well of a 96-well plate was added 100 μL of the suspension that was cultured in the presence of rHSP60 for 24 or 48 hours. To evaluate caspase-8 activity, the 24-hour cultures were incubated with 50 μM of caspase-8 inhibitor (IETD-CHO). Four hours before the end of culture, to each well was added 7 μL of 3-(4,5-dimethylthiazol-2-yl)-2,5-diphenyltetrazolium bromide (MTT, Sigma-Aldrich) at a concentration of 5 mg/mL. The plates were centrifuged at $300 \times g$, the supernatant removed, and the pellets of cells were lysed with 100 μL DMSO (Sigma-Aldrich). The plates were read at a wavelength of 570 nm. All the experiments using animals deficient of molecule were performed using the same protocols.

## Statistical analysis

Statistical differences between means in each experiment were performed using a one-way analysis of variance test (ANOVA) with subsequent post hoc test analysis using a Bonferroni correction. Unpaired two-tailed t-test was applied to determine significance between two

experimental groups. Differences were considered statistically significant with a *P* value less than 0.05.

## Results

### rHSP60 inhibits the viability and proliferation of lymphocytes

To evaluate whether rHSP60 induced an antigen-specific cell-mediated immunity, we pulsed CFSE-labeled pLN cells from rHSP60-immunized mice with rHSP60 at a concentration of 2 or 10 μg/mL for 48 hours. The cells did not respond to these stimuli (Fig 1A). Even when we tested these cells with rHSP60 at a concentration of 25 or 50 μg/mL, the results were similar to stimulation with 2 and 10 μg/mL of rHSP60. The proliferative ability of these cells did not seem to be compromised since they responded to ConA, a T-cell polyclonal stimulator (Fig 1A). These experiments were repeated with unstained cells in an MTT assay as an alternative to CFSE assay, and we observed similar results where rHSP60 decreased the viability of pLN cells by about 20% (Fig 1B). Because we had shown previously that treatment of *P. brasiliensis* infected mice with three doses of rHSP60 had a beneficial effect on the infection while a single dose worsened it [14], next, we did the MTT assay with spleen cells from mice immunized i.m. with three doses of rHSP60 at 15-day intervals. As seen in Fig 1B, the *in vitro* stimulation with rHSP60 in concentrations as low as 1.5 μg/mL diminished the viability of the cells. Remarkably, the cells stimulated with both ConA plus rHSP60 had a decrease of 60% in comparison to the group incubated only with ConA (Fig 1B), i.e., the viability loss induced by rHSP60 was seen even in spleen cell cultures co-incubated with a polyclonal inductor of T cell proliferation. When we used LPS in the cultures instead ConA, we observed that the proliferation of B lymphocytes decreased significantly, suggesting that the effect of rHSP was on lymphocytes (S1 Fig). Also, these results suggested that the decrease in cell proliferation was not related to the immunization but to the effect of rHSP60 on cells in culture. To confirm this, we evaluated the effect of rHSP60 in another assay based on OVA-specific cell-mediated immune response. As expected, mononuclear cells from pLN of OVA-immunized mice that were pulsed with OVA plus rHSP60 had decreased the number of mononuclear cells by about 45% when compared with only OVA (Fig 1C). Therefore, our data indicate that rHSP60 has a proliferation inhibitory effect, as was shown in the assays of specific or polyclonal stimuli. To determine whether rHSP60 had the same effect on cells from non-immunized mice, we incubated spleen cells from naive mice with rHSP60 for 24 hours in an MTT assay and saw a significant diminish of cell viability at concentrations of rHSP60 as low as 3 μg/mL (Fig 1D). Because nitric oxide (NO) could have this effect on the spleen cells [18], we measured it in cell cultures from naive mice stimulated with rHSP60 for 24 hours. We observed that rHSP60 did not increase the nitrite concentration, suggesting that NO is unrelated to the phenomenon (S2 Fig).

### Lymphocytes are the mainly population affected by rHSP60

Next, we investigate which immune cell subset was affected by the effect of rHSP60. We observed that the incubation of spleen cells from non-immunized mice incubated with rHSP60 at concentrations of 25 or 50 μg/mL significantly decreased the number of mononuclear cells (Fig 2A). We chose the concentration of 25 μg/mL of rHSP60 to stimulate the cultures to analyze the cell subsets that were more affected, and our data showed that both T (CD3+ cells) and B (CD19+ cells) lymphocytes had a lower number when compared to unstimulated control cell (medium) (Fig 2B–2D). Moreover, CD11b+ were not affected by the stimulus when compared with non-stimulated cells (medium) (S3 Fig).

To confirm whether rHSP60 was toxic primarily to lymphocytes, we incubated murine B-cell lymphoma cell line A20 and murine macrophage cell line J774 with different

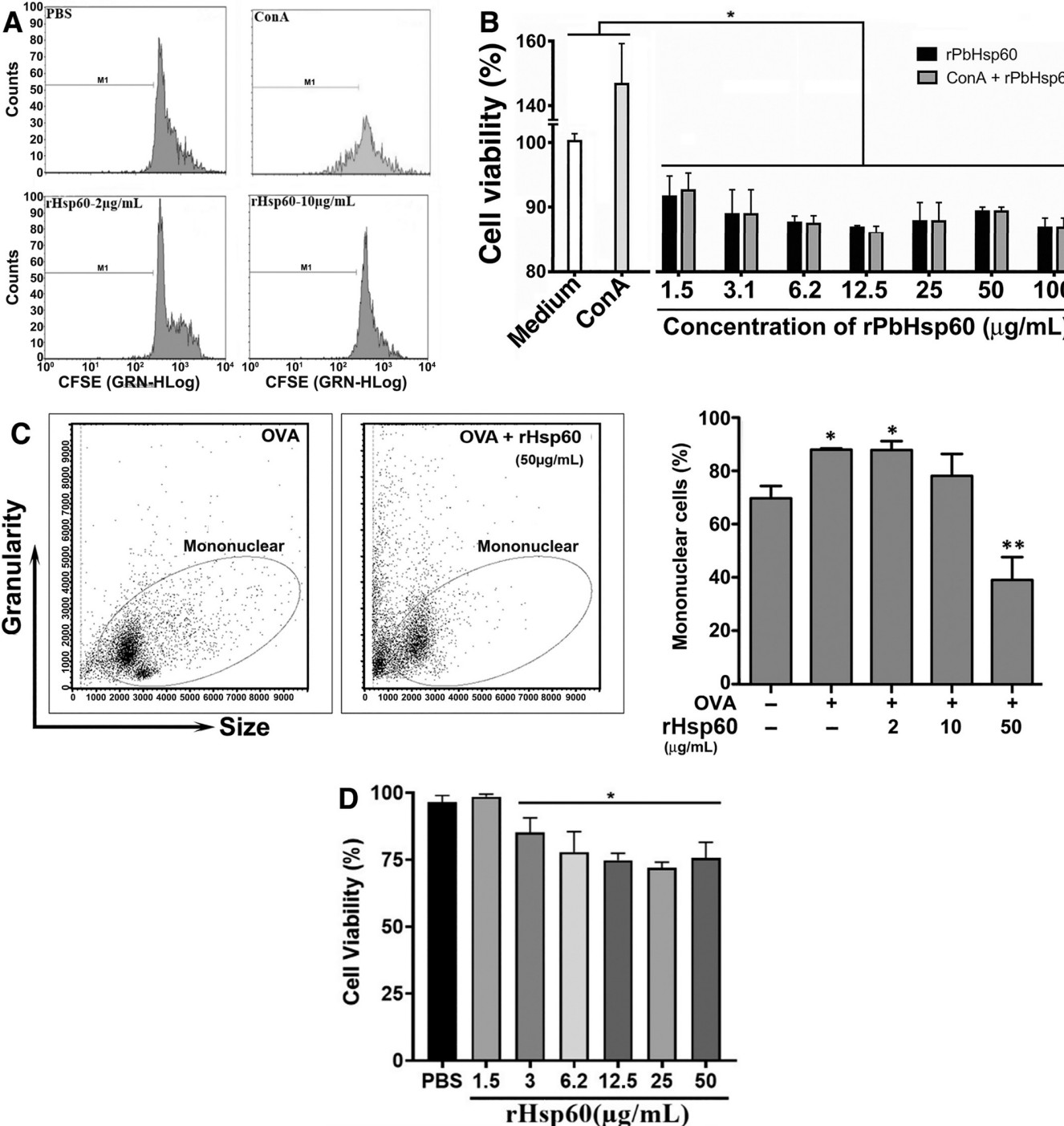

**Fig 1. rHSP60 restrains T lymphocyte proliferation by causing a reduction of spleen cell viability. A**—pLN cells ($1 \times 10^6$ cells/mL) from rHSP60-immunized mice were collected and stained with 1.25 μM of CFSE. Cell suspensions were washed and cultured in the presence of 2 or 10 μg/mL of rHSP60 for 48 hours. The negative control consisted of vehicle (PBS)-incubated cells whereas the positive one was pLN cells stimulated with 2.5 μg/mL of ConA. The cells were acquired in Guava Cytometer and analyzed in FlowJo software. **B**—Spleen cell cultures ($2 \times 10^5$ cells/mL) from WT mice immunized i.m. with 3 doses of rHSP60 were stimulated with 2.5 μg/mL of ConA in the presence or absence of different concentrations of rHSP60 for 48 hours. The cultures were also performed only with either Pb HSP60r or ConA. MTT solution was added to each well 4 hours before the end of the culture time, and the percentage of viability was obtained by reading the absorbance at 570 nm of the cell lysate. Unstimulated cells were used as a control (medium). **C**—pLN cells from mice immunized with 50 μg of OVA were labeled with CFSE and cultured without stimulus (Medium), with only OVA at 50 μg/mL or with OVA added with 2, 10, and 50 μg/mL of rHSP60 for 48 hours. Bars represent means ± SD of cell percentage. *$P < 0.05$ in relation to cultures without stimulus or stimulated with OVA plus 50 μg/mL of rHSP60. **$P < 0.05$ compared with other groups. Next to the bar graph are two graphs showing representative data from a single culture from the experiment performed

in duplicate for cultures stimulated with OVA and with OVA containing 50 µg/mL of rHSP60. **D**—Spleen cells from non-immunized mice ($2 \times 10^5$ cells/mL) were cultured in the presence of different concentrations of rHSP60. The incubation with MTT solution and analysis was done as described above to panel B. *$P < 0.05$ in relation to cultures stimulated with PBS or with rHSP60 at 1.5 µg/mL of rHSP60.

concentrations of rHSP60 and analyze their viability after 24 hour-stimulation. As observed for primary mouse spleen cell cultures, rHSP60 was toxic for lymphocyte cell line (Fig 3A), but not for macrophage one (Fig 3B).

### rHSP60 signaling through TLR4/TRIF to induce the cytotoxic effect

HSP60s are molecules highly conserved in the evolution [7] and have been related to proinflammatory effects and apoptosis through binding to TLR4 [19]. Based on these data, we supposed TLR signaling could be responsible for lymphocyte death triggered by rHSP60. Thus, we first used spleen cells from mice deficient in MyD88, an essential adaptor protein for signaling of the most TLRs, and we saw by flow cytometry that the decrease in the number of mononuclear cells induced by rHSP60 probably was not dependent on these receptors (Fig 4A).

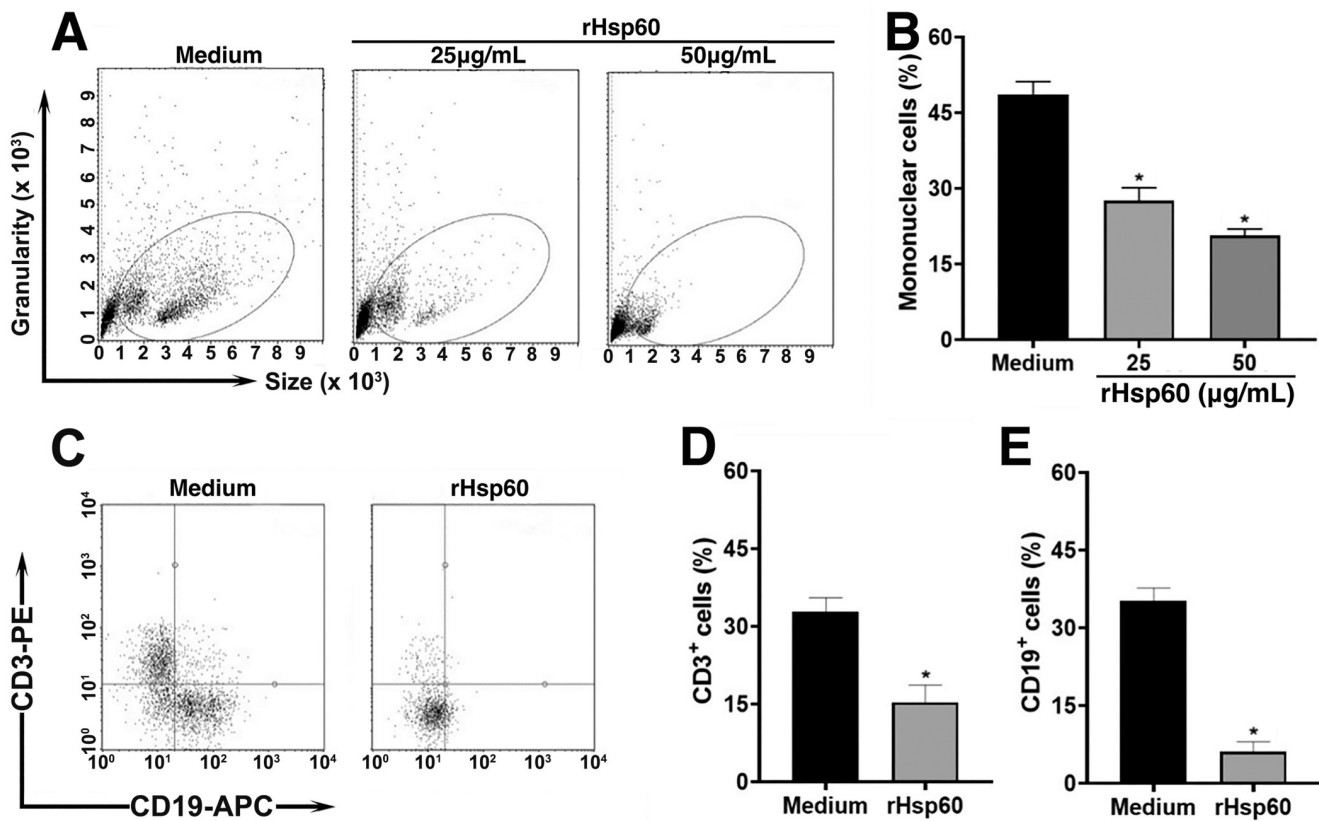

**Fig 2. Lymphocytes are the main spleen cell population affected by rHSP60. A**—Spleen cells at $4.5 \times 10^5$ cells/mL were incubated with 25 or 50 µg/mL of rHSP60 for 24 hours. The cells were acquired in the Guava cytometer and FlowJo software was used to analyze the FSC and SSC parameters. **B**—Results are presented as mean ± SEM from the number of gated mononuclear cells shown in panel A. They are representative of two different experiments in duplicate. **C**—Gated spleen mononuclear cells described above, stimulated for 24 hours in the presence or absence of 25 µg/mL of rHSP60, were labeled with anti-CD3-PE and anti-CD19-APC for 1 hour. Cell suspensions were acquired in the Guava Cytometer and data for CD3 and CD19 labeling were analyzed using FlowJo software. **D** and **E**—Graphs show mean ± SEM of (**D**) CD3+ cells and (**E**) CD19+ cells from two different experiments in duplicate. *$P < 0.05$ in relation to control (medium).

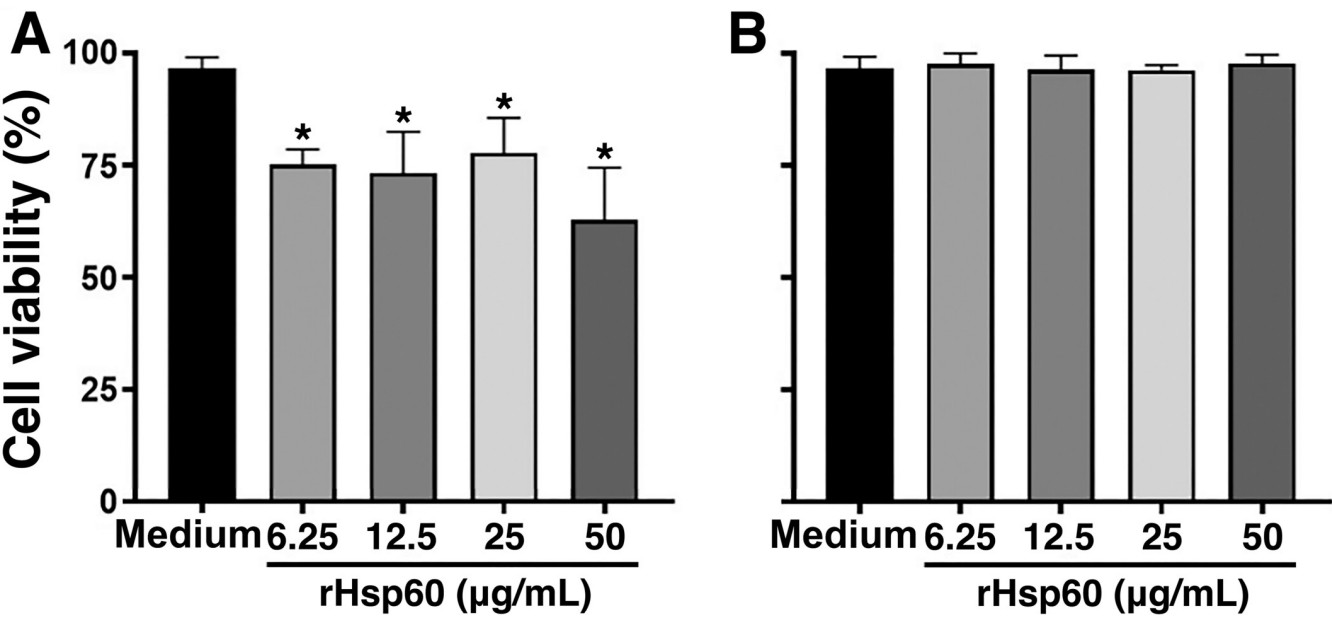

**Fig 3. rHSP60 decreases viability of B lymphocyte cell line but not macrophage one.** A suspension of $2 \times 10^5$ cells/mL of mouse cell line A20 (**A**) or J774 (**B**) was stimulated with a range of 6.25 to 50 μg/mL of rHSP60 for 24 hours. The incubation with MTT solution and analysis was done as described in Fig 1, panel B. Results represent mean ± SEM from two independent experiments in triplicate. *$P < 0.05$ in relation to control (medium).

However, we could not rule out that the effect of rHSP60 was triggered by TLR3, which signals via the adapter protein TRIF, as well as by TLR4, which signals by both MyD88 and TRIF. Unlike the spleen cells from MyD88-deficient and control WT mice that had a decrease in the number of mononuclear cells after incubation with rHSP60 (Fig 4A), TRIF-deficient spleen cells did not have their viability affected (Fig 4B). To assess whether the effect of rHSP60 was triggered via TLR3 or TLR4, we repeated these experiments with spleen cells from TLR3-/-, TLR4-/- mice. rHSP60 induced the reduction of viability on TLR3-/- cells but not on TLR4-/- (Fig 4B), suggesting that rHSP60 from *P. brasiliensis* stimulates cell death through TLR4/TRIF pathway.

In this study, we used samples of recombinant protein with low LPS contamination, and even when we added LPS to the culture, rHSP60 was able to decrease the relative quantity of cells (S1 Fig). Even so, because the phenomenon involved TLR4, we still performed two independent experiments. In the first, spleen cell cultures were stimulated with rHSP60 digested with proteinase K, which resulted in a drastic reduction in the cytotoxic effect of rHSP60 (Fig 4C). In the other, spleen cell cultures were incubated with rHSP60 in the presence of PMX at 30 μg/mL, which is able to block the effect of LPS in the production of tumor necrosis factor (TNF) by macrophages [20], but not reduce the cytotoxicity of rHSP60 (Fig 4D). Together these data suggest that the cytotoxic effect was dependent on rHSP60.

### The effect of rHSP60 on cells is dependent of TNF and its receptor 1 (TNFR1)

Previously, we have shown that infected mice treated with rHSP60 markedly increased cytokine concentrations, including TNF [15]. Because cell death by apoptosis or necroptosis via TLR4/TRIF may be associated with TNF [21, 22], we sought to evaluate the TNF role in the phenomenon of cytotoxicity. When TNF neutralizing antibody were used, we observed

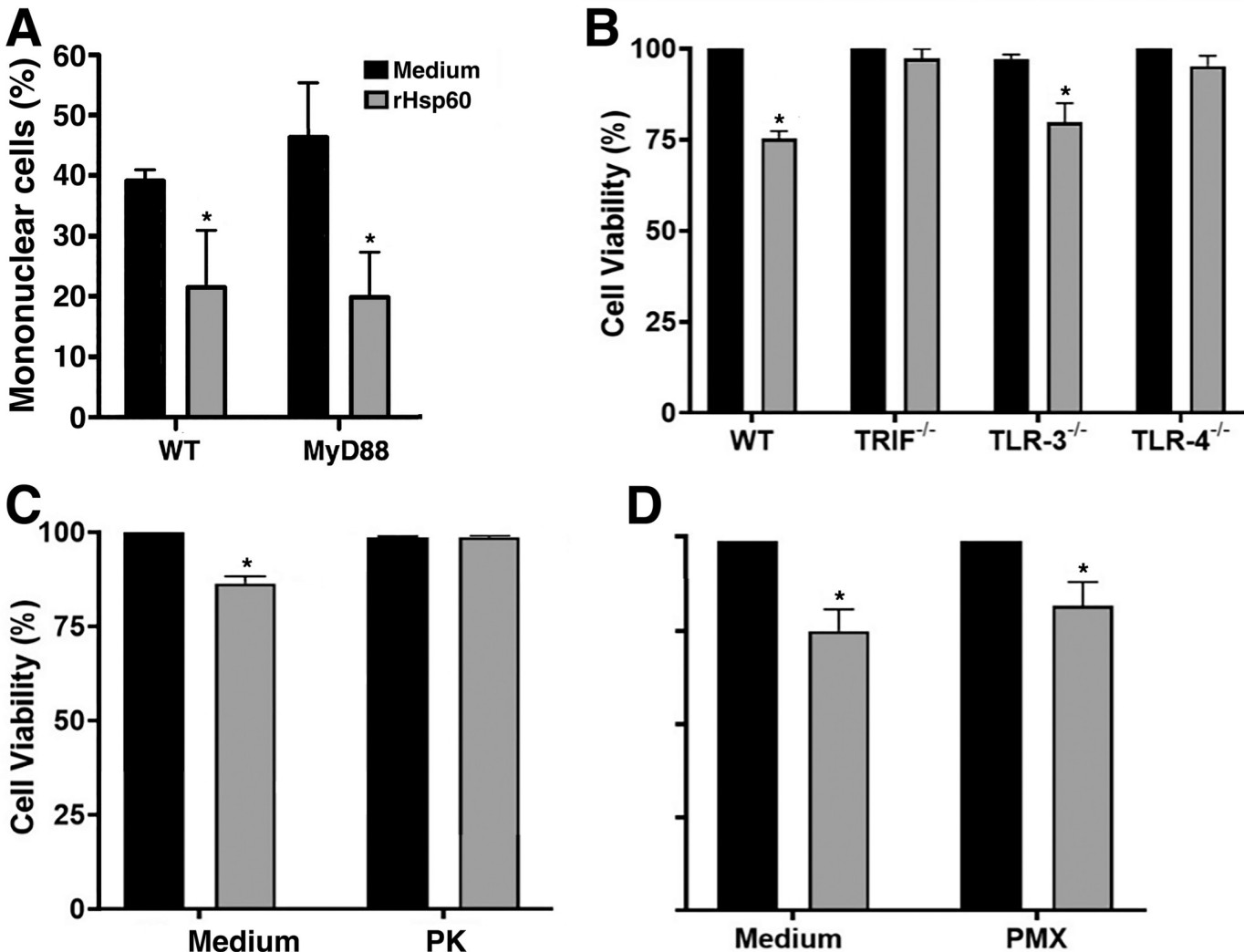

**Fig 4. Cytotoxicity effect induced by rHSP60 is via TLR4/TRIF. A**—Spleen cells from MyD88[-/-] and WT mice were stimulated with 50 μg/mL of rHSP60 for 24 hours, labeled with anti-CD3-PE and anti-CD19-APC antibodies and subjected to flow cytometry. The cells were gated and analyzed as show in Fig 2. The results are representative data of a single culture of the experiment performed in duplicate for cultures. *$P < 0.05$ in comparison with the cultures of unstimulated spleen cells (medium). **B**—Culture of spleen cells from WT mice or mice deficient in TLR3 (TLR3[-/-]), TLR4 (TLR4[-/-]) or TRIF (TRIF[lps/lps]) were stimulated with rHSP60 for 24 hours. Results represent mean ± SEM from two different experiments in triplicate. *$P < 0.05$ in comparison with the cultures of unstimulated spleen cells (medium). **C**—Spleen cells from naive mice were stimulated with rHSP60 digested or not with proteinase K (PK) for 24 hours. Controls were cells stimulated only with PK or without stimulation. *$P < 0.05$ in comparison with the other groups. **D**—Spleen cells from WT mice were stimulated with rHSP60 treated or not with polymyxin-B (PMX) for 24 hours. *$P < 0.05$ in comparison with the cultures of unstimulated spleen cells (medium). In panel, **B**, **C**, and **D**, MTT solution was added to each well 4 hours before the end of the culture time. The percentage of viability was obtained by reading the absorbance at 570 nm of the cell lysate. Unstimulated cells were used as a control (medium).

blocking in the loss of cell viability when compared to cells treated only with rHSP60 (Fig 5A). To confirm this result, we used spleen cells from TNFR1[-/-] or TNFR1R2[-/-] mice, and consistent with the TNF antibody neutralization experiments, we observed spleen cells from both TNF receptor-deficient mice, in contrast to spleen cells from WT mice, did not have viability loss when stimulated with rHSP60 (Fig 5B).

The decrease in the viability and number of spleen mononuclear cells, probably lympho-cytes, induced by rHSP60 associated with TLR4/TRIF and TNFR1 signaling led us to suggest that rHSP60 could be inducing apoptosis in the cells. Indeed, spleen cells cultured with 25 μg/

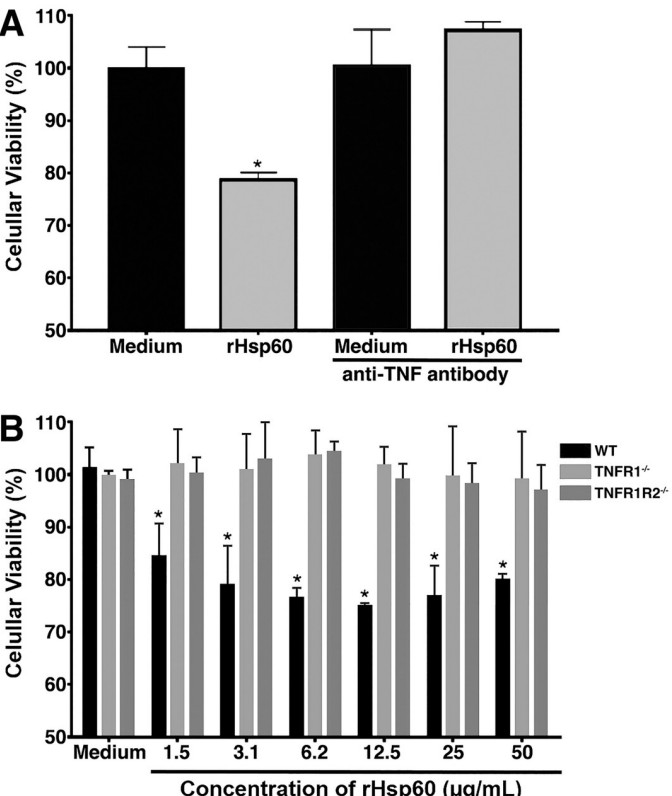

**Fig 5. Cell death is induced by an indirect mechanism dependent on TNF and TNFR1. A**–Spleen cells from WT mice were stimulated for 24 hours with 50 μg/mL of rHSP60r in the absence or presence of anti-TNF antibodies. *$P < 0.05$ in comparison with the other groups. **B**—Spleen cells from WT, TNFR1$^{-/-}$ or TNFR1R2$^{-/-}$ mice were stimulated for 24 hours with different concentrations of rHSP60. *$P < 0.05$ in comparison with unstimulated spleen cells (medium) from all mice and spleen cells stimulated with rHSP60 from deficient mice. MTT solution was added to each well 4 hours before the end of the culture time. The percentage of viability was obtained by reading the absorbance at 570 nm of the cell lysate. Unstimulated cells were used as a control (medium). Data are representative two independent experiments.

mL of rHSP60 for 24 hours increased the number cells labeled with annexin V and PI, probably corresponding to late apoptosis (Fig 6A). Because is know that TNFR1 is one of the receptors whose signaling initiates the extrinsic apoptosis pathway [22, 23], so we evaluated whether the inhibition of caspase-8 and consequently of this pathway could avoid cell death triggered by rHSP60. When spleen cell cultures were treated with the caspase-8 inhibitor IETD-CHO and stimulated with 25 μg/mL of rHSP60, we observed that these cells did not have their viability decreased when compared to control cultures not stimulated with rHSP60, in contrast to cultures treated only rHSP60 (Fig 6B). The above results suggest that HSP60 from *P. brasiliensis* induces lymphocyte death in a mechanism dependent on TLR4/TRIF and TNFR1 signaling with the consequent induction of the extrinsic apoptosis pathway.

## Discussion

In the present work, we found that rHSP60 from *P. brasiliensis* can induce cell death of spleen lymphocytes by apoptosis. Our results suggest that the apoptosis and, consequently, the reduction in spleen cells induced by rHSP60 involve TLR4/TRIF and TNFR signaling on a mechanism dependent on caspase-8.

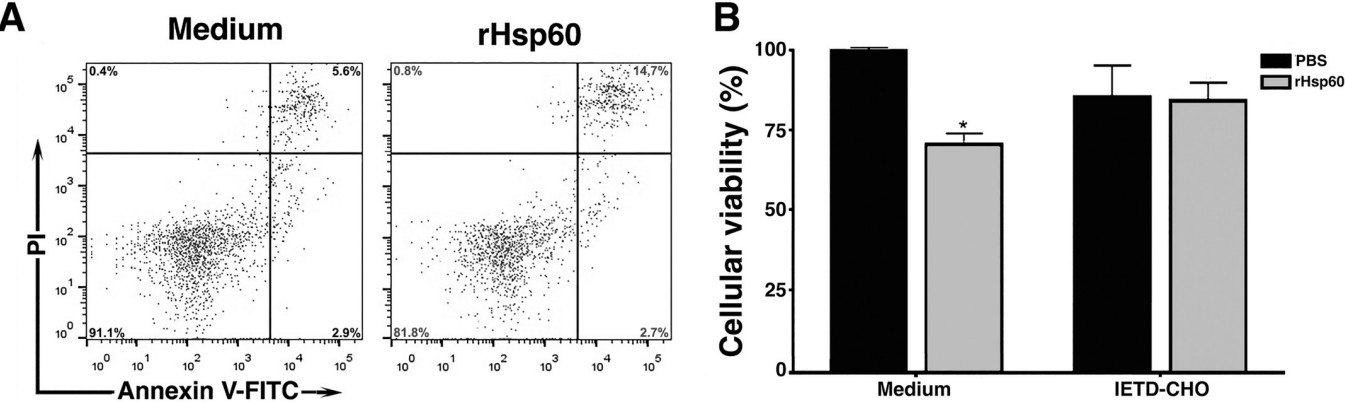

**Fig 6. rHSP60-induced apoptosis of spleen cells is mediated by caspase-8. A**—Culture of spleen cells ($2.5 \times 10^5$ cells/mL) were incubated in the absence (Medium) or presence of with 25 μg/mL of rHSP60. The cell suspensions were labeled with Annexin-V FITC and PI. About 10,000 cells of each sample were acquired, gated according to forward and side scatter parameters. After acquisition in FACSCanto II Flow cytometer, the data were analyzing using FlowJo software. These results are representative of three independent experiments. **B**—Spleen cells from WT mice were stimulated with 25 μg/mL of rHSP60 in absence (Medium) or presence of 50 μM of caspase-8 inhibitor (IETD-CHO) for 24 hours. MTT solution was added to each well 4 hours before the end of the culture time, and the percentage of viability was obtained by reading the absorbance at 570 nm of the cell lysate. Unstimulated cells were used as a control (medium). Results represent mean ± SD from two different experiments in triplicate. *$P < 0.05$ in comparison with the other groups.

Pathogens can modulate the immune system to survive and multiply in their hosts. The release of antigens as secreted immunomodulators is one of the strategies used by microbes, such as intracellular ones [24], to change the immune equilibrium in their favor, which includes *Paracoccidioides* fungi [25]. Our group has shown that rHSP60 can cause deleterious effects in mice treated with one application of rHSP60 without showing the precise mechanism [15], in contrast to protection developed when used in 3-dose schedules of vaccination [17] or therapy [15]. This HSP60 action was so robust that it reversed the beneficial therapy effect of the complete Freund adjuvant on experimental PCM [15]. Thus, it is not surprising that when a monoclonal antibody against anti- HSP60 from *Histoplasma capsulatum*, which is highly homolog to *Paracoccidioides* HSP60, was used to treat experimental PCM, protected the infected mice [26], perhaps by neutralizing of the effect of HSP60. Initially, we thought that maybe HSP60 could contribute to a scenario of immune imbalance and, therefore, a harmful immune response [27]. We evaluated the cell-mediated immunity against rHSP60 in a classical procedure where cells from the drain pLN of mice immunized in the footpad with rHSP60 were stimulated, *in vitro*, with rHSP60. Surprisingly, we observed that rHSP60 incubation rendered undivided cells seen by the CFSE cell proliferation assay.

This lack of proliferation in cells stimulated with rHSP60 could be explained by the production of high concentrations of nitric oxide (NO) in the cultures because this mediator is known to suppress lymphocyte replication, as reviewed by Napoli et al. [28]. This phenomenon was also seen in experimental PCM, as spleen cells from infected mice had an augment of proliferation when nitroarginine, an inhibitor of nitric oxide synthase (NOS), was given to the animals [29]. Other authors found similar results in which spleen cells from *P. brasiliensis*-infected mice presented a suppression of the ConA-induced proliferation in a manner dependent on NO concentrations. So, these cells treated with L-NMMA, an inhibitor of inducible NOS, had this suppression of cell proliferation reverted [18]. Moreover, there was no significant increase or decrease in the number of cells in cultures in which the ConA-stimulated cell proliferation was suppressed by NO compared to the non-stimulated control [18, 29]. Here, we did not detect significant differences between the nitrite concentrations produced by

rHSP60-treated spleen cells and those released by unstimulated cells (medium) (S2 Fig), suggesting that the absence of proliferation was not due to NO. Even when we used drain pLN cells from OVA-immunized mice, the cells incubated with OVA plus rHSP60 lost viability instead of replicating as the positive control of the cells incubated with only OVA. The influence of rHSP60 in spleen cells was independent of the source of mice since cells of non-immunized and immunized mice do not proliferate and have viability loss.

Interestingly, our analysis showed that the loss of cell viability occurred mainly in the non-adherent cell populations, which comprised mainly lymphocytes. To confirm these results, we evaluated which population in spleen cells decreased after stimulation with rHSP60, i.e., both T (CD3+ cells) and B (CD19+ cells) lymphocytes had a lower number when compared to unstimulated control (medium) (Fig 2) while CD11b+, in contrast, was not affected by the stimulus (S3 Fig).

Toll-like receptors are studied mainly due to their role in innate immune cells and inflammation [30]. However, they also signal in adaptive immune cells and can cause activation [31, 32] or cell death [33]. We decided then considered a premise about the loss of cell viability triggered by HSP60 from effects described by others in which HSP60s stimulate proinflammatory effects and apoptosis through binding to TLR4 [19, 34] even that previously one of us has observed that spleen cells deficient in MyD88, an adaptor protein to TLR4 signaling, could undergo the rHSP60 effect (Fernandes F.F., personal communication). Noticeably, when we evaluated rHSP60 in spleen cells from TLR4-/- mice, we observed the reversal of loss of cell viability. Because TLR4 can signal via two pathways through adaptor MyD88 or TRIF, we investigated whether rHSP60 induced TLR4 signaling in a manner dependent on the adaptor TRIF once we discarded the effect by MyD88. Indeed, there was no loss of viability in rHSP60-stimulated spleen cells from TRIF-/- mice.

The discovery that the loss of viability of spleen cells stimulated by rHSP60 depended on TLR4/TRIF signaling allowed us to study possible mechanisms related to this signaling pathway [35], of which TNF could have an essential role [36]. Significantly, anti-TNF antibodies use inhibited the rHSP60 action in decreasing the number of splenic cells. Moreover, we observed that the spleen cells from animals deficient in the TNFR1 also did not undergo the death effect of rHSP60, suggesting that the signaling triggered by rHSP60 was TNF and TNFR1 dependent. These findings were interesting because the TNF binding to TNFR1 can cause cell survival, apoptosis, or a regulated form of necrosis, necroptosis [37], outcomes that depend on the balance achieved between pro- and anti-apoptotic signals [38]. To cell death induced by TNFR1 signaling, the receptor-interacting serine/threonine-protein kinase 1 (RIPK1) was described as essential in leading to apoptosis or necroptosis [39]. The RIPK1 activation in TNF-stimulated cells regulates a transient multimeric complex, which is associated with the cytosolic domain of TNFR1. The ubiquitination of RIPK1 in the K63 site chain promotes the formation of complex I and induces pro-survival signaling by activation of transcription factors, such as NF-κB, while the deubiquitination mediates the formation of complex II by binding of RIPK1 with Fas-associated death domain and caspase-8, with subsequent apoptosis [40, 41]. In our experiments, the loss of cell viability induced by rHSP60 was due to the splenic lymphocyte apoptosis and linked to the activation of caspase 8, given that the caspase-8 inhibitor IETD-CHO prevented the action triggered by rHSP60. We suggested apoptosis because, with the caspase-8 inhibition, there would be recruitment of RIPK3 into complex II that would activate necroptosis [40, 42].

In aging, T lymphocyte subsets and their stages of activation may show a variation in sensitivity to TNF-induced cell death [43]. Add to that the infectious agents, such as *Plasmodium vivax*, which induce a possible evasion mechanism of the immune system through apoptosis of T lymphocytes indirectly by activating TNFR1 [44]. Indeed, there are still knowledge gaps that

require further investigation about the effect of TNF on infectious processes, one of which is how pathogens, such as *Paracoccidioides* fungi, can manipulate the cytokine signaling and modify the fate of adaptive immune cells. Although additional data are required to corroborate the idea that rHSP60 plays a direct role in the immunopathology of paracoccidioidomycosis, we suggest that rHSP60 reduces the proliferation and causes cell death of lymphocytes, which might contribute to suppression of the immune response in PCM.

In summary, our data show that rHSP60 from *P. brasiliensis* induces cell death in murine lymphocytes depending on TLR4, TRIF adaptor protein, and TNFR1. Besides, the experiments allow us to suggest that the cell death effects are dependent on caspase-8 activation, as the use of a specific caspase inhibitor was able to reverse the toxic effect of rHSP60. Therefore, we propose that TLR4 signaling via TRIF to TNF production, in turn, induces activation of caspase-8 and apoptosis.

## Supporting information

**S1 Fig. rHSP60 decreases LPS-induced mononuclear cell proliferation.** Spleen cells ($4.5 \times 10^5$ cells/mL) from WT mice were stained with CFSE and stimulated with only LPS at 25 μg/mL (0) or LPS plus 2, 10, or 50 μg/mL of rHSP60 for 24 hours. The negative control consisted of unstimulated cells (medium). The cells were acquired in a Guava Cytometer and analyzed in FlowJo software. Bars represent means ± SD of cell percentage. $^*P < 0.05$ compared with unstimulated cells (Medium) and cells stimulated with rHSP60 at 50 μg/mL. $^{**}P < 0.05$ compared with other groups. Experiment representative of two experiments.
(TIF)

**S2 Fig. rHSP60 did not stimulate nitrite production by spleen cells.** Spleen cells ($4.5 \times 10^5$ cells/mL) from WT mice were stimulated with 25 μg/mL of rHSP60 for 24 hours in the presence or absence of 30 μg/mL of polymyxin. The negative control consisted of unstimulated cells (medium). The supernatants were collected and the $NO_2^-$ determined with Griess reagent. Bars represent means ± SD of $NO_2^-$ concentrations. Experiment representative of two experiments.
(TIF)

**S3 Fig. CD11b$^+$ spleen cells are not affected by the stimulus with rHSP60.** Adherent spleen cells ($4.5 \times 10^5$ cells/mL) from WT mice were stimulated with rHSP60 at 25 or 50 μg/mL for 24 hours. The negative control consisted of unstimulated cells (Medium). The cells were labeled with PerCP-Cy5.5-conjugated anti-CD11b antibodies, acquired in a Guava Cytometer, and analyzed in FlowJo software. Bars represent means ± SD of cell percentage. Experiment representative of two experiments.
(TIF)

## Acknowledgments

The authors thank Mrs. Denise B. Ferraz and Dr. Taise N. Landgraf for their excellent technical assistance in cytometry, and Dr. João Santana da Silva from the Oswaldo Cruz Foundation, Ribeirão Preto, who provided us with A20 and J774 cell lines.

## Author Contributions

**Conceptualization:** Ademilson Panunto-Castelo.

**Data curation:** Igor Emiliano L. Souza, Fabrício F. Fernandes.

**Formal analysis:** Igor Emiliano L. Souza, Fabrício F. Fernandes, Ademilson Panunto-Castelo.

**Investigation:** Igor Emiliano L. Souza, Fabrício F. Fernandes, Ademilson Panunto-Castelo.

**Methodology:** Igor Emiliano L. Souza, Fabrício F. Fernandes.

**Project administration:** Ademilson Panunto-Castelo.

**Supervision:** Ademilson Panunto-Castelo.

**Validation:** Igor Emiliano L. Souza, Ademilson Panunto-Castelo.

**Visualization:** Igor Emiliano L. Souza.

**Writing – original draft:** Igor Emiliano L. Souza, Fabrício F. Fernandes, Ademilson Panunto-Castelo.

**Writing – review & editing:** Ademilson Panunto-Castelo.

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
