## [Decision Letter · Decision Letter 0]

2 Feb 2024

PONE-D-23-42226Recombinant 60-kDa heat shock protein from Paracoccidioides brasiliensis induces the death of mouse lymphocytes in a mechanism dependent on Toll-like receptor 4 and tumor necrosis factor.PLOS ONE

Dear Dr. Panunto-Castelo,

Thank you for submitting your manuscript to PLOS ONE. After careful consideration, we feel that it has merit but does not fully meet PLOS ONE’s publication criteria as it currently stands. Therefore, we invite you to submit a revised version of the manuscript that addresses the points raised during the review process.

We look forward to receiving your revised manuscript.

Kind regards,

Aristóbolo M Silva

Academic Editor

PLOS ONE

“The São Paulo Research Foundation [grant numbers 2017/01390-8, and 2021/01528-5], and CAPES.”

“The authors thank Mrs. Denise B. Ferraz and Dr. Taise N. Landgraf for their excellent technical assistance in cytometry, and Dr. João Santana da Silva from the Oswaldo Cruz Foundation, Ribeirão Preto, who provided us with A20 and J774 cell lines. This work was supported by the São Paulo Research Foundation – FAPESP [grant numbers 2017/01390-8, and 2021/01528-5].”

“The São Paulo Research Foundation [grant numbers 2017/01390-8, and 2021/01528-5], and CAPES.”

Reviewers' comments:

Reviewer's Responses to Questions

**Comments to the Author**

1. Is the manuscript technically sound, and do the data support the conclusions?

Reviewer #1: Yes

Reviewer #2: Yes

2. Has the statistical analysis been performed appropriately and rigorously? 

Reviewer #1: Yes

Reviewer #2: Yes

3. Have the authors made all data underlying the findings in their manuscript fully available?

Reviewer #1: No

Reviewer #2: Yes

4. Is the manuscript presented in an intelligible fashion and written in standard English?

Reviewer #1: Yes

Reviewer #2: Yes

5. Review Comments to the Author

Reviewer #1: The manuscript entitled "Recombinant 60-kDa heat shock protein from Paracoccidioides brasiliensis induces the death of mouse lymphocytes in a mechanism dependent on Toll-like receptor 4 and tumor necrosis factor.", by Souza, I et al., provides valuable information on the role of Hsp60 in the pathogenesis of experimental Paracoccidiodomycosis. My major concern regards the great number of "Data not shown" in the manuscript. These data should be provided as part of the manuscript or its supporting information.

Reviewer #2: In the present study, the authors evaluate the occurrence of apoptosis induced by rHSP60 in a mechanism dependent of TLR4/ TRIF/TNFR1 pathway. To this, the proliferation of spleen cells and of drain pLN after rHSP60 challenged was evaluated as well as cell viability and apoptosis on spleen cells. As result the data show that rHSP60 from P. brasiliensis induces cell death in murine lymphocytes in a manner depending on TLR4, TRIF adaptor protein via TNFR1 with the production of TNF, and caspase-8 activation. As conclusion the findings suggest that rHSP60 from P. brasiliensis reduces the proliferation and causes cell death of lymphocytes, which might contribute to suppression of the immune response in PCM. The data are very interestingly, however, minor issues need to be addressed.

Page 3, line 49: Please correct the sentence “…the lungs.t cases registered, about 85%, held in Brazil…”

Page 7, lines 161 and 162: Please correct the sentence “…last dose spleen of animals was euthanized and had the spleen removed.”

Page 8, line 185: Please correct the formation of the word “and” in the title “rHSP60 inhibits the viability and proliferation of lymphocytes”

Legend of figure 2, page 11, line 252: Please put in bold the letter “B” in the sentence “B - Results are presented as mean…”

Page 12, line 271: Please explain whether the group control is the same of cells treated with medium only. If they are the same please uniformize the name of the groups in the legends and graphics.

Legend of figure 5, page 14: Please inform the number of replications in this experiment.

Figures 1A, 2A, and C, 6A: Please increase the quality of them. The words are blurred.

Figure 1B: Please correct the numbers at the Y-axis, the number 100 appears twice.

6. PLOS authors have the option to publish the peer review history of their article (what does this mean?). If published, this will include your full peer review and any attached files.

Reviewer #1: No

Reviewer #2: No

---

## [Author Response · Author response to Decision Letter 0]

13 Feb 2024

Response to reviewers

Reviewer #1:

My major concern regards the great number of "Data not shown" in the manuscript. These data should be provided as part of the manuscript or its supporting information.

Answer: Thank you. As asked, we changed the manuscript, providing some data and modifying the text to become clearer.

On page 9, lines 200-202, the data not shown were very similar to the one shown, so we modified the following.

“Even when we tested these cells with rHSP60 at a concentration of 25 or 50 μg/mL, the results were similar to stimulation with 2 and 10 μg/mL of rHSP60.”

On page 9, line 206, we think that Fig 1B could replace the data not shown.

On page 9, lines 216, and page 13, line 306, we added a supporting figure (S1 Fig).

On pages 11, lines 261, and page 18, line 425, we added a supporting figure (S3 Fig).

On page 13, line 311, “data not shown” was replaced by a reference.

On page 17, line 414, we added a supporting figure (S2 Fig), which was also added to a new sentence in the results (on page 10, lines 228-229).

Reviewer #2:

Page 3, line 49: Please correct the sentence “…the lungs.t cases registered, about 85%, held in Brazil…”

Answer: Thank you. The mistake has been fixed. 

… the lungs, and about 85% of registered cases have occurred in Brazil [1-3].

Page 7, lines 161 and 162: Please correct the sentence “…last dose spleen of animals was euthanized and had the spleen removed.” 

Answer: Thank you. The mistake has been fixed. Now, on page 8, lines 173 and 174.

“Seven days after the only or last dose of rHSP60, the animals were euthanized as described above and had the spleen removed.”

Page 8, line 185: Please correct the formation of the word “and” in the title “rHSP60 inhibits the viability and proliferation of lymphocytes”

Answer: Thank you. The mistake has been fixed. Now, on page 9, line 197.

Legend of figure 2, page 11, line 252: Please put in bold the letter “B” in the sentence “B Results are presented as mean…”

Answer: Thank you. The mistake has been fixed. Now, on page 11, line 266.

Page 12, line 271: Please explain whether the group control is the same of cells treated with medium only. If they are the same please uniformize the name of the groups in the legends and graphics.

Answer: Thank you. We reviewed all the text. Now, on page 12, line 285.

Legend of figure 5, page 14: Please inform the number of replications in this experiment. Answer: Thank you. The number of replications was shown. Now, on page 15, line 352.

Figures 1A, 2A, and C, 6A: Please increase the quality of them. The words are blurred.

Answer: Thank you. We improved the figure quality.

Figure 1B: Please correct the numbers at the Y-axis, the number 100 appears twice. 

Answer: Thank you. The mistake was corrected.

---

## [Decision Letter · Decision Letter 1]

27 Feb 2024

Recombinant 60-kDa heat shock protein from Paracoccidioides brasiliensis induces the death of mouse lymphocytes in a mechanism dependent on Toll-like receptor 4 and tumor necrosis factor.

PONE-D-23-42226R1

Dear Dr. Panunto-Castelo,

We’re pleased to inform you that your manuscript has been judged scientifically suitable for publication and will be formally accepted for publication once it meets all outstanding technical requirements.

Kind regards,

Aristóbolo M Silva

Academic Editor

PLOS ONE

Additional Editor Comments (optional):

Reviewers' comments:

Reviewer's Responses to Questions

**Comments to the Author**

1. If the authors have adequately addressed your comments raised in a previous round of review and you feel that this manuscript is now acceptable for publication, you may indicate that here to bypass the “Comments to the Author” section, enter your conflict of interest statement in the “Confidential to Editor” section, and submit your "Accept" recommendation.

Reviewer #1: All comments have been addressed

Reviewer #2: All comments have been addressed

2. Is the manuscript technically sound, and do the data support the conclusions?

Reviewer #1: Yes

Reviewer #2: Yes

3. Has the statistical analysis been performed appropriately and rigorously? 

Reviewer #1: Yes

Reviewer #2: Yes

4. Have the authors made all data underlying the findings in their manuscript fully available?

Reviewer #1: Yes

Reviewer #2: Yes

5. Is the manuscript presented in an intelligible fashion and written in standard English?

Reviewer #1: Yes

Reviewer #2: Yes

6. Review Comments to the Author

Reviewer #1: (No Response)

Reviewer #2: In the present study, the authors evaluate the occurrence of apoptosis induced by rHSP60 in a mechanism dependent of TLR4/ TRIF/TNFR1 pathway. To this, the proliferation of spleen cells and of drain pLN after rHSP60 challenged was evaluated as well as cell viability and apoptosis on spleen cells. As result the data show that rHSP60 from P. brasiliensis induces cell death in murine lymphocytes in a manner depending on TLR4, TRIF adaptor protein via TNFR1 with the production of TNF, and caspase-8 activation. As conclusion the findings suggest that rHSP60 from P. brasiliensis reduces the proliferation and causes cell death of lymphocytes, which might contribute to suppression of the immune response in PCM.

The authors have addressed and explained all questions raised by the previous reviewed.

7. PLOS authors have the option to publish the peer review history of their article (what does this mean?). If published, this will include your full peer review and any attached files.

Reviewer #1: No

Reviewer #2: No
